# A Pilot Study on Remote Sensing and Citizen Science for Archaeological Prospection

**Christopher Stewart [1],*** , **Georges Labrèche [1]** and **Daniel Lombraña González [2]**

1    European Space Agency, Earth Observation Programmes Directorate, Future Systems Department, 00044 Frascati, Italy; georges.labreche@esa.int
2    Scifabric Ltd., Southampton SO15 3FG, UK; daniel@scifabric.com
*    Correspondence: chris.stewart@esa.int

**Abstract:** Cost-effective techniques for systematic archaeological prospection are essential to improve the efficiency of preventive archaeology and the preservation of cultural heritage. Web Mapping Services, such as Microsoft Bing Maps, provide imagery covering extensive areas at high resolution. These can, in some cases, reveal cropmarks of buried historical structures. Given that archaeological prospection is not generally the priority of most common Web Mapping Services, the conditions under which images are acquired are not always suitable for the appearance of cropmarks. Therefore, their detection is typically serendipitous. This pilot project attempts to assess the potential to use the Microsoft Bing Maps Bird's Eye service within a crowdsourcing platform to systematically search for archaeological cropmarks in the surroundings of the city of Rome in Italy. On this platform, which is hosted by the company Scifabric (Southampton, UK) and based on PyBossa, an Open Source framework for crowdsourcing, members of the public are invited to interpret oblique air photo tiles of Bing Maps Bird's Eye. While the project is still on-going, at least one seamless coverage of tiles in the area of interest has been interpreted. For each tile, the Bing Maps Bird's Eye service provides oblique air photo coverage in up to four possible orientations. As of 5 July 2020, 18,765 of the total 67,014 tasks have been completed. Amongst these completed tasks, positive detections of cropmarks were recorded once for 1447 tasks, twice for 57 tasks, and three or more times for 10 tasks. While many of these detections may be erroneous, some correspond with archaeological cropmarks of buried remains of buildings, roads, aqueducts, and urban areas from the Roman period, as verified by comparison with archaeological survey data. This leads to the conclusion that the Bing Maps Bird's Eye service contains a wealth of information useful for archaeological prospection, and that to a certain extent citizen researchers could help to mine this information. However, a more thorough analysis would need to be carried out on possible false negatives and biases related to the varying ease of interpretation of residues of different archaeological structures from multiple historical periods. This activity forms the first part of a research project on the systematic prospection of archaeological cropmarks. The ultimate aim is to reach a critical mass of training data through crowdsourcing which can be augmented and used as input to train a machine learning algorithm for automatic detection on a larger scale.

**Keywords:** archaeology; prospection; crowdsourcing; citizen science; cropmarks; remote sensing; Bing Maps; earth observation; Rome

## 1. Introduction

### 1.1. Archaeological Prospection

Particularly in countries with a rich cultural heritage, the abundance of buried archaeological structures poses a challenge. The preservation of the archaeological record is often a conflicting requirement with territorial and economic development [1]. This has led to the concepts of preventive archaeology and rescue archaeology as pre-emptive attempts to survey evidence of former actions of humankind before they are replaced by fresh actions [2]. To maximise the preservation of cultural heritage with the least impact on development, efficient and cost-effective techniques are needed for archaeological prospection. Timely and accurate information on the location of any unknown or partially known archaeological sites of value enables better planning of any development or preservation activities [1–3].

### 1.2. Remote Sensing for Archaeological Prospection

While ground-based methods for archaeological prospection can provide accurate information on buried archaeological structures, even in a non-invasive manner with techniques such as geophysical prospecting (e.g., magnetometry and ground-penetrating radar [4–6]), when applied to large areas they rapidly become costly and time-consuming. Remote sensing data from Unmanned Aerial Vehicle (UAV), aircraft, or satellite platforms have the obvious added value in their ability to rapidly survey extensive areas, with a synoptic view of features which may not be as evident from a ground perspective [7]. A number of active and passive remote sensing methods have become established for archaeological prospection. UAV or airborne Light Detection And Ranging (LiDAR) can detect subtle topographic traces of buried structures [8–10], even through dense forest cover [8,11]. Synthetic Aperture Radar (SAR), through its sensitivity to surface roughness and the relative permittivity of materials [12], has enabled direct detection of structures buried beneath a cover of sand [13,14], and indirect detection of buried objects through surface proxies, including differential vegetation growth (cropmarks), moisture anomalies, and topographic traces [15,16].

Since the beginning of the 20th century, passive remote sensing in the form of air photography has been used extensively to detect archaeological residues [17–19]. The use of wavelengths outside the spectrum of visible light has been shown to improve the detection of certain residues: the high reflection of near-infrared radiation by the cell structure of healthy green plants [20] is exploited to better detect cropmarks [21,22]. Thermal infrared remote sensing has been used to detect buried archaeological structures, such as through analysis of thermal inertia [23,24]. Other studies have demonstrated the utility of other parts of the spectrum, such as passive UV imaging [25]. Much research has been carried out on indices and algorithms involving multiple wavelengths in multispectral [26–28] and hyperspectral [29,30] imagery to better discriminate archaeological residues.

Aside from spectral characteristics, acquisition time is another key element in remote sensing for archaeological prospection. Certain archaeological residues can be notoriously ephemeral. Cropmarks, for example, may only appear for short periods [31] and under particular circumstances, such as during periods of drought [32,33] or at other times [34]. For the best visibility of archaeological residues, the season and time of day are also relevant. The angle of the sun, for example, may affect the visibility of topographic residues by casting shadows (shadow marks) [17]. The pointing angle of the sensor is a further element that may affect the clarity of residues in remote sensing data. Imagery acquired at an oblique angle in a particular orientation may in some cases better reveal residues, but the choice of orientation is key [17,35].

### 1.3. Data Availability and WMS

In the age of Big Data and in the new paradigm of "Newspace", we are seeing a reduction in costs and a dramatic increase in data volume, velocity of acquisition, and spectral variety of remotely sensed data [36,37]. Nonetheless, Very High Resolution (VHR) data acquisition campaigns are still costly,

particularly if large areas are required to be covered. However, much archived remote sensing data exists that is freely available, which in some cases can provide near seamless coverage of extensive areas. The VHR optical data provided by Web Mapping Services (WMS) such as Google Maps, Google Earth, Bing Maps and Apple Maps constitute excellent resources for many applications, and on numerous occasions have been applied to archaeological prospection, e.g., [38–40].

The drawback of such popular services is that access to the raw data is usually not provided, thus greatly limiting the processing that can be applied to the data. While some services do provide a sophisticated processing environment, such as Google Earth Engine, they usually do not include free access to raw VHR data, only visualisations of such data, or full access to lower resolution imagery such as from the Sentinel and Landsat satellite missions. The clarity of archaeological residues can be greatly enhanced with many processing techniques, such as involving the use of filters, indices, data fusion methods, object detection algorithms and machine learning classifiers [14,21,26,27,41]. Without access to the raw data, most of these cannot be applied [42].

Nonetheless, while processing capability may not always be offered, many WMS do provide free access to visualisations of a large quantity of VHR data, in some cases multiple images for each area. Examples include the time slider of Google Earth and the Microsoft Bing Maps Bird's Eye service, which provides visualisations of oblique air photos acquired from up to four directions. While the Bird's Eye service only covers selected regions, such as major cities and surrounding areas, the air photos provide very high resolution, which is necessary to distinguish cropmarks of certain archaeological structures such as the walls of Roman-era buildings. If systematically studied, these may contain a wealth of information on the location of potential buried archaeological structures, particularly in areas rich in cultural heritage, such as in the outskirts of Rome, Italy.

However, the imagery on most of these popular platforms was not acquired specifically for archaeologists, and their utility is generally serendipitous. If an image happens to have been acquired during conditions favourable to their appearance, archaeological cropmarks may be visible over a given area (e.g., Figure 1). The utility of these platforms for archaeological survey is therefore limited and often considered unscientific and opportunistic. Systematic application over extensive areas is inefficient and time-consuming, given the low rate of success and the need for human interpretation.

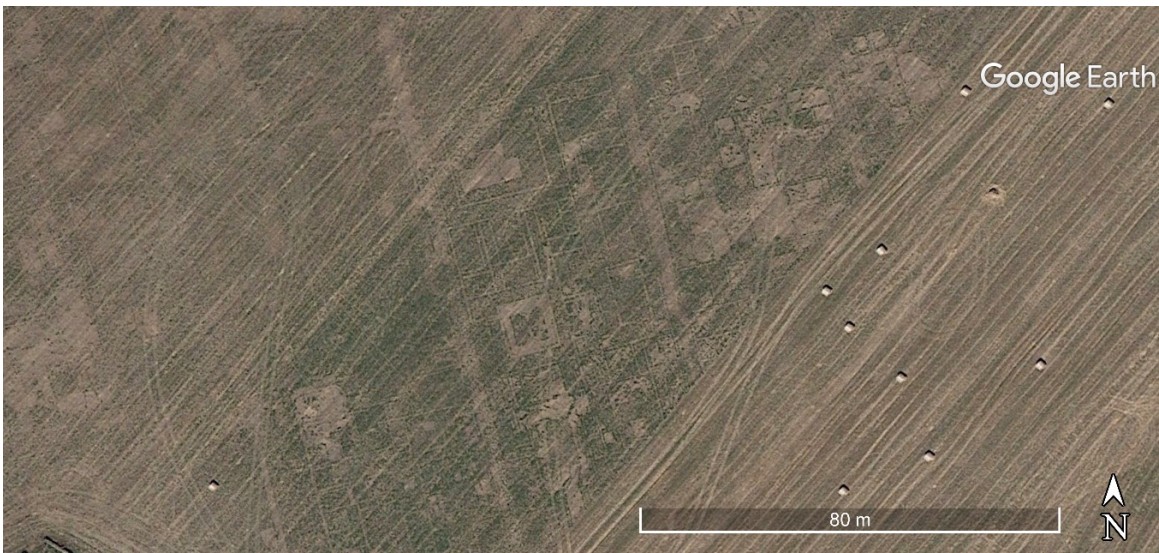

**Figure 1.** An example of cropmarks in Google Earth in the case of the ancient Roman city of Ferentium. The image acquisition date is reported to be 7 August 2013. Courtesy of Google Earth Pro.

*1.4. Citizen Science for Archaeological Prospection*

The uses of citizen science and "community archaeology" are well-established in cultural heritage [43,44]. Their application for direct archaeological interpretation using remotely sensed

imagery is less common, but has recently begun to attract interest. An example of such an activity is a crowdsourcing project, launched by a team of researchers, together with National Geographic, to search for the tomb of Genghis Khan. Volunteers were asked to tag known features and structures, as well as those they thought could reveal ancient finds [45]. Another example includes GlobalXplorer, launched in 2017, which seeks to identify and quantify looting and encroachment to sites of archaeological and historical importance using crowdsourcing, with a secondary objective to discover new archaeological sites [46]. The combination of both crowdsourcing and machine learning has been applied to archaeological prospection in the Netherlands with LiDAR data, where various communities of citizen researchers were engaged in both remote sensing data interpretation and fieldwork [47].

### 1.5. Introduction to the Pilot Study

The project outlined here is a pilot activity that builds on some previous results of community engagement for archaeological prospection. It aims to determine to what extent the efficiency of using the freely available Bird's Eye service of Microsoft Bing Maps for archaeological prospection can be improved by addressing the challenges of human interpretation through citizen science. The purpose of the project is not to attempt a rigorous archaeological survey of the area of interest, but merely to better understand the potential of the Bird's Eye service to provide information about the presence of any cropmarks in the culturally rich area surrounding the city of Rome, and the capability of citizen researchers to retrieve these. A platform for citizen science was created at the European Space Agency (ESA), Frascati, Italy, which uses a template of PyBossa, an Open Source framework for crowdsourcing, created and hosted by the company Scifabric. This platform is used by citizen researchers to classify Bird's Eye tiles of Microsoft Big Maps according to whether or not they contain archaeological cropmarks. This classification is just a simple binary choice between yes (the tiles shown contain cropmarks) or no (no cropmarks are evident). If successful, and if a sufficient number of cropmarks can be retrieved, a secondary aim would be to assess the possibility of using these to train a machine-learning algorithm to carry out systematic prospection, similar to [47–49]. Given the complexity of cropmarks, and the consequent need for significant amounts of training data, considerable data augmentation would likely be required to increase the available training sample. The citizen science project is still incomplete, and therefore this paper presents only the preliminary results of the first phase of the project involving the use of citizen researchers to extract cropmarks using the Bird's Eye service of Bing Maps.

## 2. Materials and Methods

### 2.1. Study Area

The Area of Interest (AOI) includes the city of Rome and its surroundings in an area of approximately $1677\,\text{km}^2$ (see Figure 2). This area has experienced extensive modern urban development, particularly in the periods of 1881–1886, 1926–1931 and 1951–1960 [50], leading to the loss of much cultural heritage [51]. Despite this, there is still a lot of undeveloped land with a high density of archaeological structures, both upstanding and buried beneath the ground, as testified by many surveys that have been carried out in various parts of this area, e.g., [51–54]. It was decided not to mask out fully urbanized areas given that many parks and large gardens exist which may contain cropmarks. If, on the other hand, a citizen researcher is confronted with a tile entirely covered by urban areas, it can be quickly classified as an area of no cropmarks.

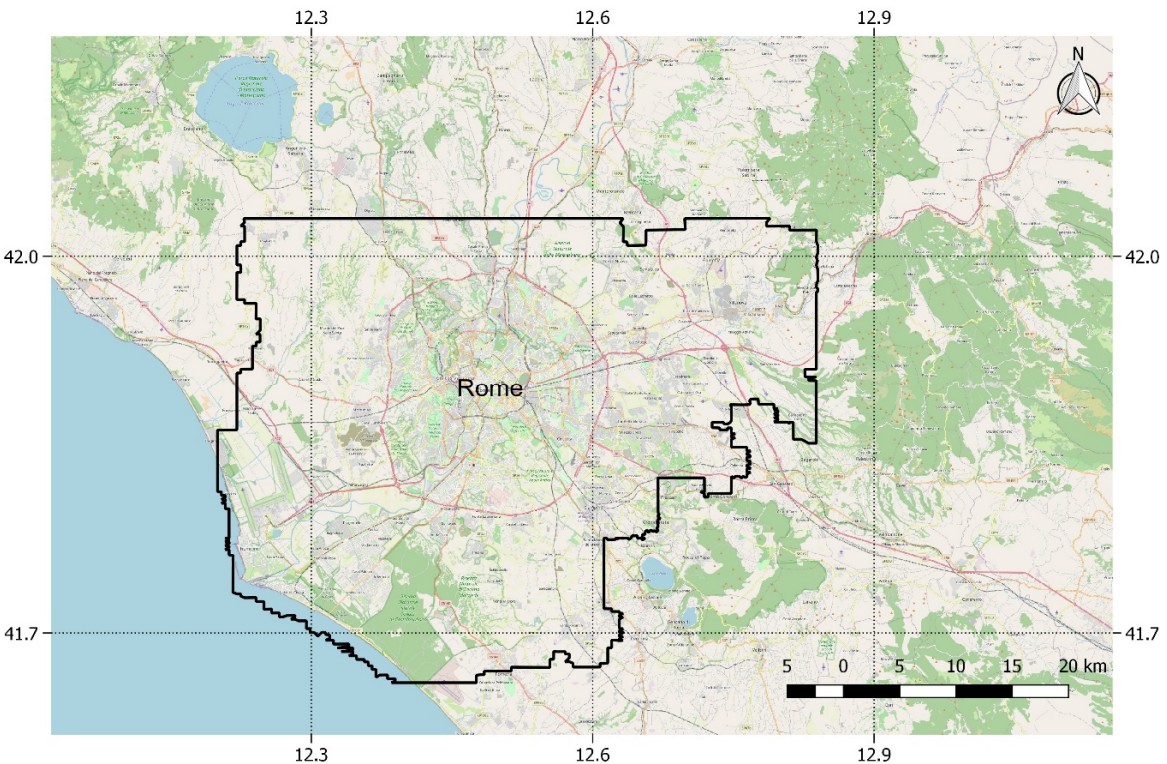

**Figure 2.** A map showing the coverage (in black outline) of Bing Maps Bird's Eye air photos used in the project, in WGS 84/Pseudo Mercator projection. Courtesy of Open Street Map.

*2.2. Method*

The project was set up by first defining the tasks to be completed. A website was then created with a user interface for carrying out each task, and some background information about the project. This website was initially developed and hosted on a crowdsourcing service called Crowdcrafting, but after discontinuation of this service in April 2019, it was migrated to an ESA website. A task run navigator was developed to extract results from the completed tasks. Figure 3 is a diagram showing the methodology.

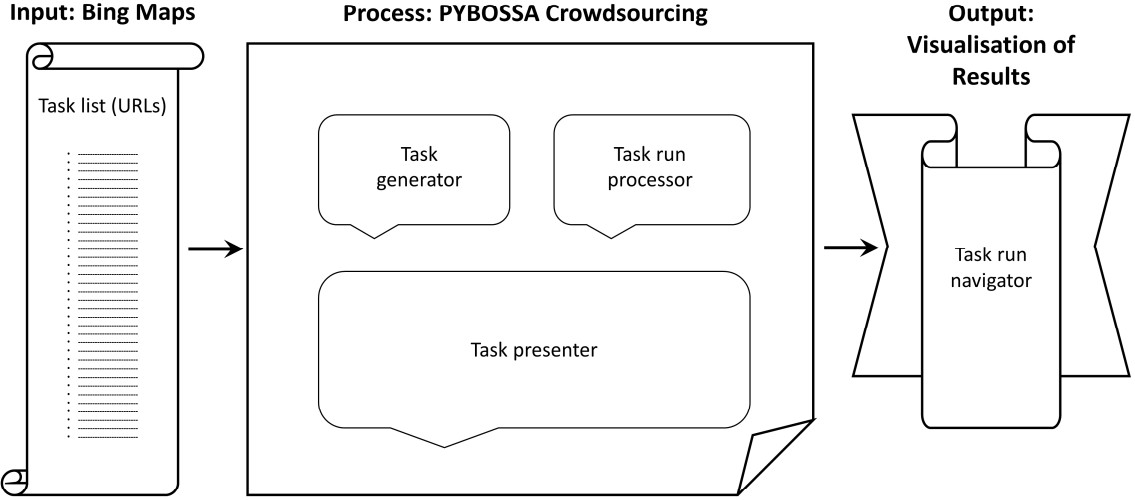

**Figure 3.** Diagram showing the various parts of the methodology, beginning with the list of URLs of Bing Maps Bird's Eye tiles as input, to the PyBossa crowdsourcing implementation, and ending with the analysis of results.

2.2.1. Task Definition

The task was defined as the binary classification of six to twelve adjacent Birds' Eye image tiles, according to whether or not an archaeological cropmark exists in any of the images. The variable number of tiles at each task depended on how many were required to fill a geographic area of 500 by 500 m. Over any one area, there are up to four different air photos acquired from four different angles (North, East, South, and West). The task redundancy was initially set to three, but later changed to one to increase the speed of interpretation in an attempt to achieve at least one full spatial coverage of the AOI. Considering the fact that each area is covered by up to four images, and hence four separate tasks, it is tempting to consider this a form of redundancy. However, not all areas are covered by four separate images. Moreover, the images were acquired at different times and orientations, and the same cropmarks may not be equally visible in all (e.g., Figure 4). There was some overlap of tiles between tasks. In many cases, for example, the bottom row of tiles in a particular task may form the top row in another task if the longitudinal increment was not sufficient to skip to another tile set.

The total number of tasks was 67,014. The Uniform Resource Locator (URL) of each of the Birds' Eye image tiles on Bing Maps to be classified was listed in a Comma Separated Values (CSV) file. For each task, a user would visit the URL of the particular image tile to be classified. An Application Programming Interface (API) key was provided by Microsoft Bing Maps for the project. This allowed 50,000 transactions (API calls) per day, which was deemed sufficient for the purposes of the project.

The tasks were initially prioritized with a "depth first" scheduling, i.e., when deciding which task to present to a user, priority was given to tasks which had already some task runs (responses), but not yet to the required redundancy of three. However, after migration of the project to the ESA website, the task priority was changed to "breadth first", which gave priority to tasks which had the least number of task runs excluding the current user. This was selected to ensure human interpretation of as wide a geographic area as possible, even if no redundancy was achieved.

2.2.2. PyBossa Implementation

Once the tasks were defined, a website was created to host the crowdsourcing project. Initially, this was developed on the free popular crowdsourcing platform Crowdcrafting of the company Scifabric. Based on PyBossa, an Open Source framework for crowdsourcing, Crowdcrafting provided templates to enable users to create crowdsourcing projects [55]. These templates included various Python scripts: a "Task Generator" is a Python script that produced a CSV file in which each row corresponds to a microtask [56]. This CSV file was imported into a PyBossa project instance in order to create the tasks within that platform. Completed tasks, called "Task Runs", were downloaded from PyBossa as CSV files. The "Task Run Processor" is another Python script that processed these Task Run CSV files into results files, also in CSV [56]. These result files grouped the task results based on the number of positive answers, i.e., all tasks with one, two, and three or more positive answers.

The "Task Presenter" is an HTML/CSS/Javascript file which described the user interface that presents the task to the crowdsourced users [56]. This contained an introduction page introducing the project: describing briefly the challenges of cultural heritage preservation, the need for archaeological prospection, a quick explanation of the mechanism of cropmarks (see Figure 5), and how they appear in optical remote sensing data, with examples. Users were invited to login, or had the option to continue anonymously, and then participate in the project.

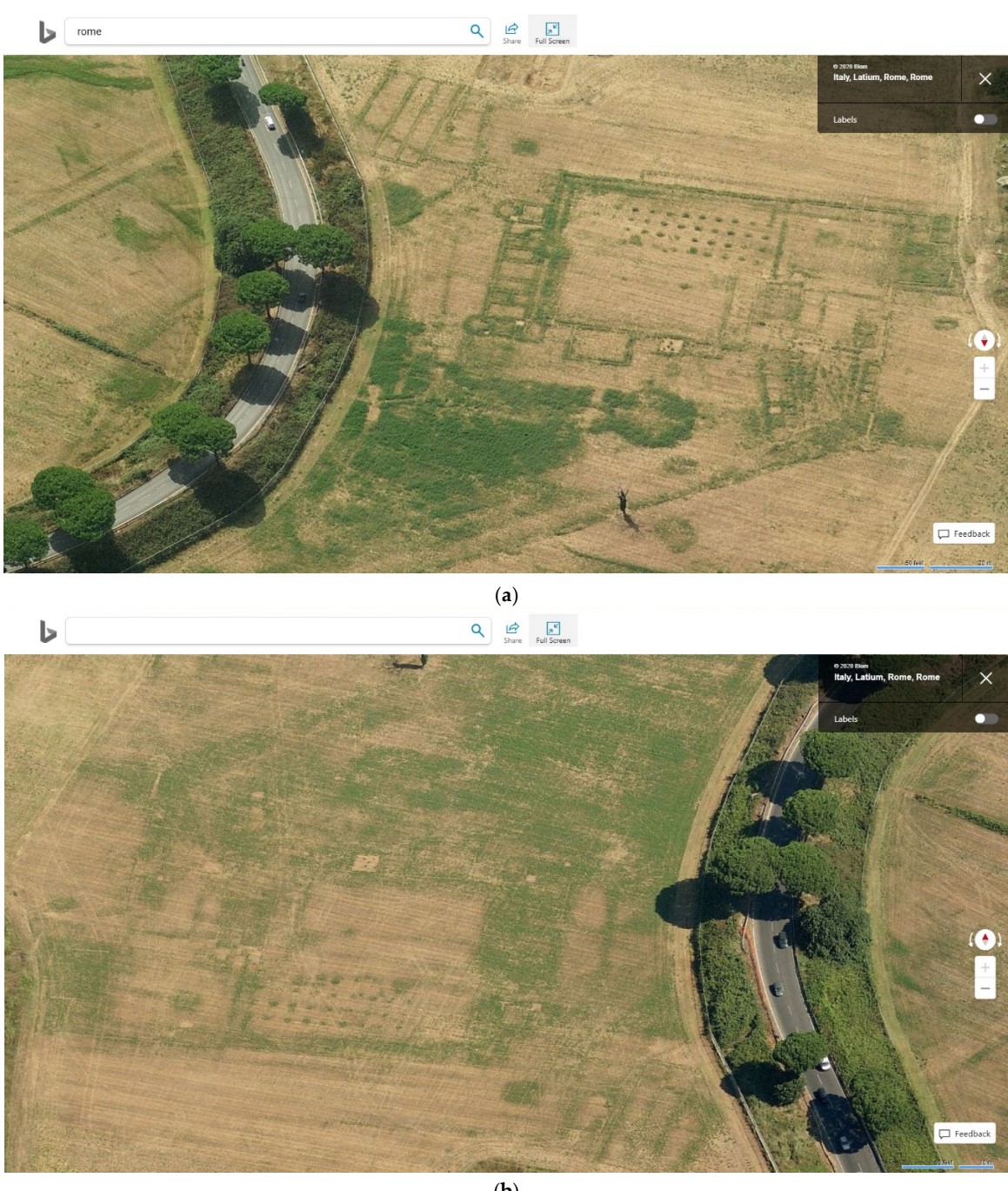

**Figure 4.** Cropmarks of archaeological structures in the area of the ancient Roman city of Ostia. The difference in the clarity of these cropmarks is apparent between the images acquired at different times (demonstrating the ephemeral nature of cropmarks) and orientations: one facing South (**a**), and the other facing North (**b**). Microsoft product screenshots reprinted with permission from Microsoft Corporation.

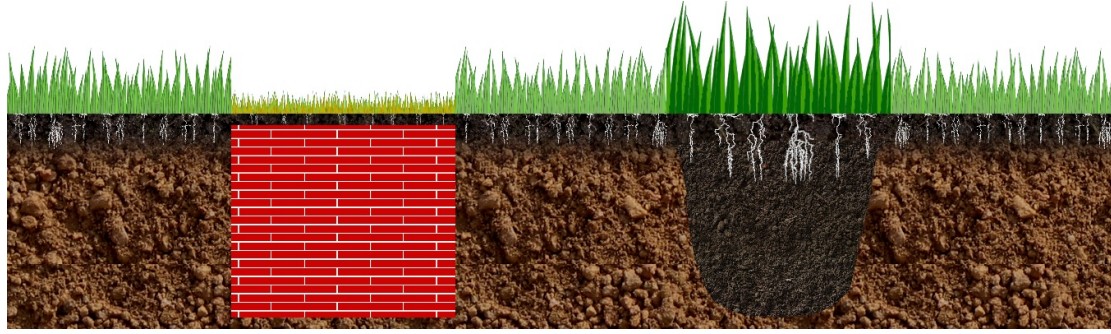

**Figure 5.** A diagram inserted in "Task Presenter" showing the physical mechanism of negative (left) and positive (right) cropmarks. The image is a modified version of the figure published by Stewart, 2017 [15].

Once users decided to contribute, they were presented with six to twelve adjacent Bing Maps Bird's Eye tiles, covering a geographic area of at least 500 by 500 m, and asked the question, "Can you see archaeological cropmarks?" Hovering the mouse icon above any of the tiles would bring up the zoom of that particular tile to enable closer inspection. Below the image tiles users could select one of three responses including, "Yes", "No", or "Bad Image". To remind users of how to interpret archaeological cropmarks, a link at the top of the same page showed examples of cropmarks overlying various archaeological structures. Once a user had selected a response, they were presented with the next task. Figure 6 is a screenshot of a task presented to a user.

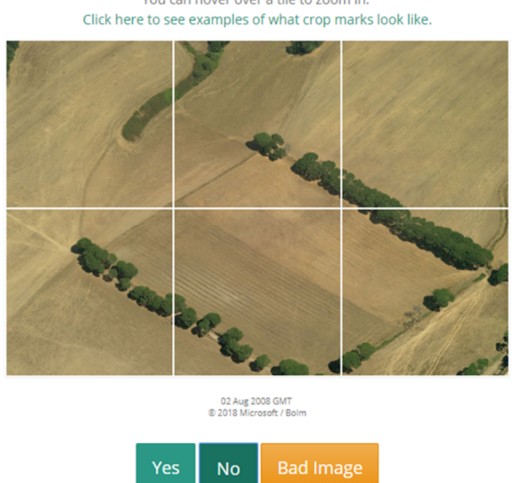

**Figure 6.** A screenshot of a task presented to a user. Microsoft products are reprinted with permission from Microsoft Corporation.

A user could also see the progress of the project by accessing a page on statistics, which contained information on the number of tasks completed, and the distribution of completed tasks over time.

In April 2019, Crowdcrafting ceased to exist. The website was therefore transferred to an Earth Observation specific crowdsourcing service hosted by Scifabric under an ESA contract [57].

2.2.3. Task Run Navigator

To extract meaningful results from the user submissions, a "Task Run Navigator" tool was developed [56] (not to be confused with the "Task Run Processor" mentioned above). This is a simple HTML page to visually navigate through the task results and filter them based on the number of detections (all tasks with one, two, and three or more positive answers).

## 3. Results

Of the 67,014 tasks, as of 5 July 2020, 18,765 have been completed. This corresponds to 28 percent of all tasks and includes at least one full coverage of the AOI with imagery in at least one orientation. Table 1 shows the number of tasks completed listed according to how many positive answers were received for each. This is shown for both the depth first scheduling, where each task was prioritised to be completed by at least three users, and breadth first (following migration to the ESA website), where each task was prioritised to be completed at least once.

**Table 1.** Tasks completed as of 5 July 2020. Different distributions of positive responses are evident according to the two types of priority tasking: depth first and breadth first.

| Positive Answers | Tasks Completed as of 5 July 2020 | | |
| --- | --- | --- | --- |
| | Crowdcrafting (Depth First, Redundancy 3) | ESA Website (Breadth First, No Redundancy) | Total Tasks |
| 0 | 787 | 16,464 | 17,251 |
| 1 | 222 | 1225 | 1447 |
| 2 | 40 | 17 | 57 |
| 3 | 10 | 0 | 10 |
| Total | 1059 (272 positive) | 17,706 (1242 positive) | 18,765 (1514 positive) |

Of the ten tasks where three positive answers were received, seven were clearly archaeological cropmarks. These included two roads; a part of an urban area; and several isolated building foundations. One of the two roads was a part of the ancient Via Salaria (see Figure 7), as identified by Quilici & Quilici Gigli [53], Hyppönen [58] and Stewart [15]. The other was a road lined with various buildings near the ancient Etruscan city of Veii, identified through comparison with a geophysical survey and interpretation carried out by Campana [54]. The urban area included parallel streets and building foundations (see Figure 8), which correspond to a part of Veii, interpreted through a comparison of the same survey and analysis of Campana [54]. The isolated building foundations were in the proximity to known archaeological structures. The remaining three of the ten tasks with three positive answers were cropmarks of man-made features, but difficult to interpret.

Of the 57 tasks with two positive responses, a fraction of these (roughly around 12 percent) corresponded with archaeological cropmarks. These included building foundations (such as that shown in Figure 9), various mixed cropmarks of natural (e.g., former rivers) and man-made structures. The remainder included cropmarks of natural features or possible man-made objects that were difficult to interpret. There were also many cases where two users classified tasks as positive but these were likely to be misinterpretations of disturbances in vegetation patterns due to alterations at the surface.

Of the 1447 tasks classified with only one positive answer, there were a number of archaeological cropmarks (corresponding with results of other surveys), or cropmarks that had the appearance of archaeological structures, even if unverified by other data. These included the same type of features as described above, but also other structures, such as aqueducts and canals from the Roman period. Figure 10, for example, shows an archaeological complex in proximity to the hexagonal port of the Roman emperor Trajan. Through comparison with an extensive magnetometry survey and interpretation published by Keay and his colleagues [52], this complex can be interpreted as an aqueduct and gravel road, running parallel to each other, and with various constructions alongside the road [52]. The classifications with one positive answer that were considered erroneous included cropmarks of modern features, such as pipelines; natural features, such as roddons (dried raised beds of watercourses); or were likely to be misinterpretations of parched and flattened grass, probably resulting from modern human actions on the surface.

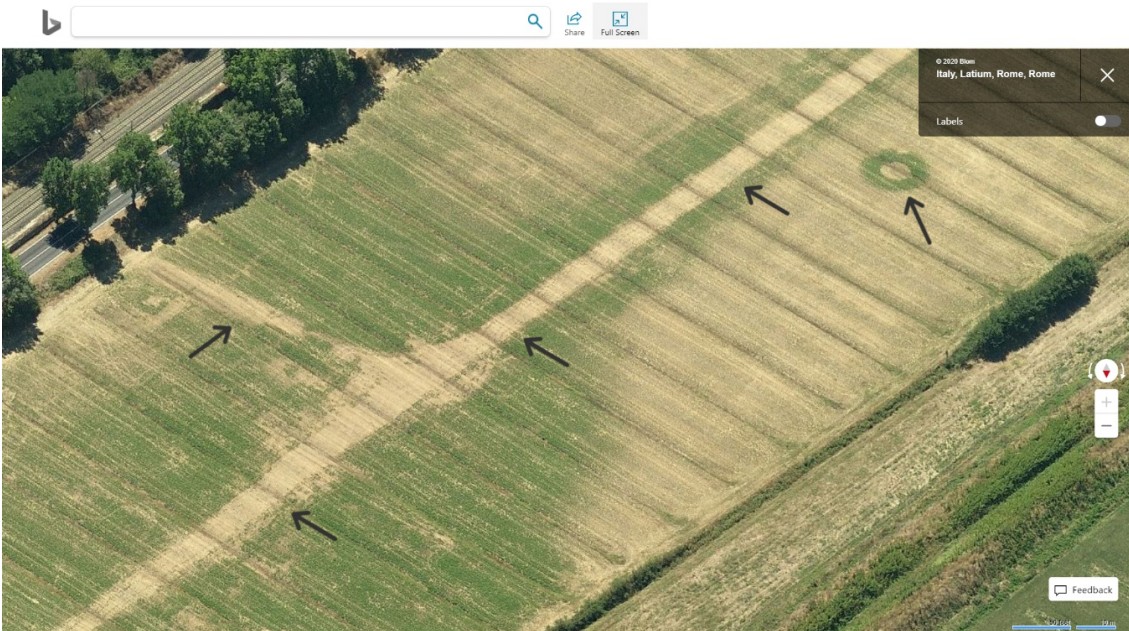

**Figure 7.** Example of archaeological cropmarks with three positive detections. The light-coloured strip (negative cropmark) bordered by darker positive cropmarks, traversing diagonally from upper right to lower left, is a part of the ancient Via Salaria, with a side road and other structures (see black arrows). These have been reported by Quilici & Quilici Gigli, 1980 [53] and Hyppönen, 2014 [58] from other optical remote sensing data, and by Stewart, 2017 [15] from SAR data. The Microsoft product screenshot is reprinted with permission from Microsoft Corporation.

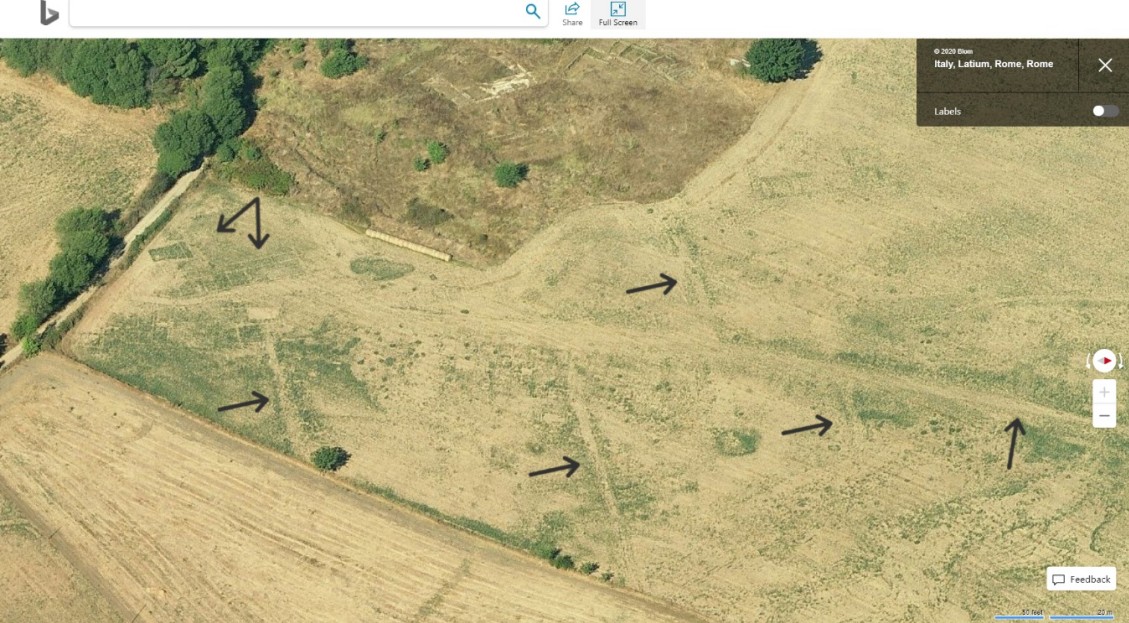

**Figure 8.** Example of archaeological cropmarks with three positive detections in the proximity of excavated structures of the ancient Etruscan city of Veii. This area has been much studied, e.g., by Ward-Perkins [59], and has been surveyed using geophysical prospection (magnetometry) by Campana [54]. Parallel lines running approximately North to South, and a wider line running East to West, have also been identified in the magnetometry survey of Campana, and have been interpreted as streets of the ancient city of Veii [54], while other cropmarks in between the streets show building foundations (see black arrows). The Microsoft product screenshot is reprinted with permission from Microsoft Corporation.

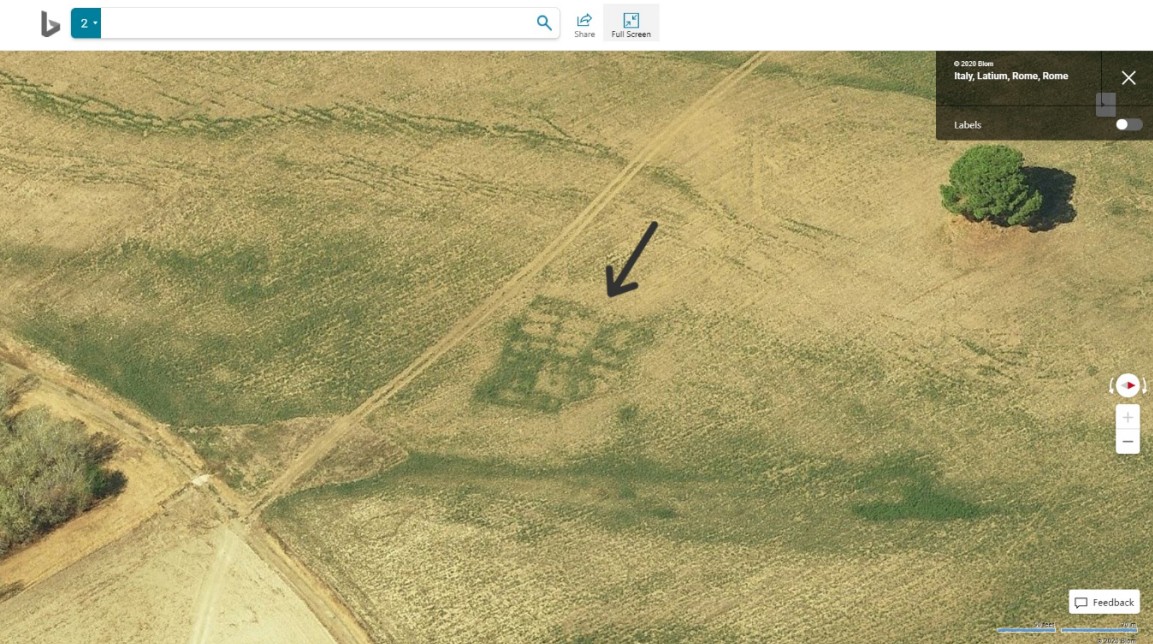

**Figure 9.** An example of an archaeological cropmark with two positive detections (see black arrow). This is in proximity to the Via Nomentana. It appears to be a cropmark of a Roman villa, but more data would be needed to verify the nature of the structure. The Microsoft product screenshot is reprinted with permission from Microsoft Corporation.

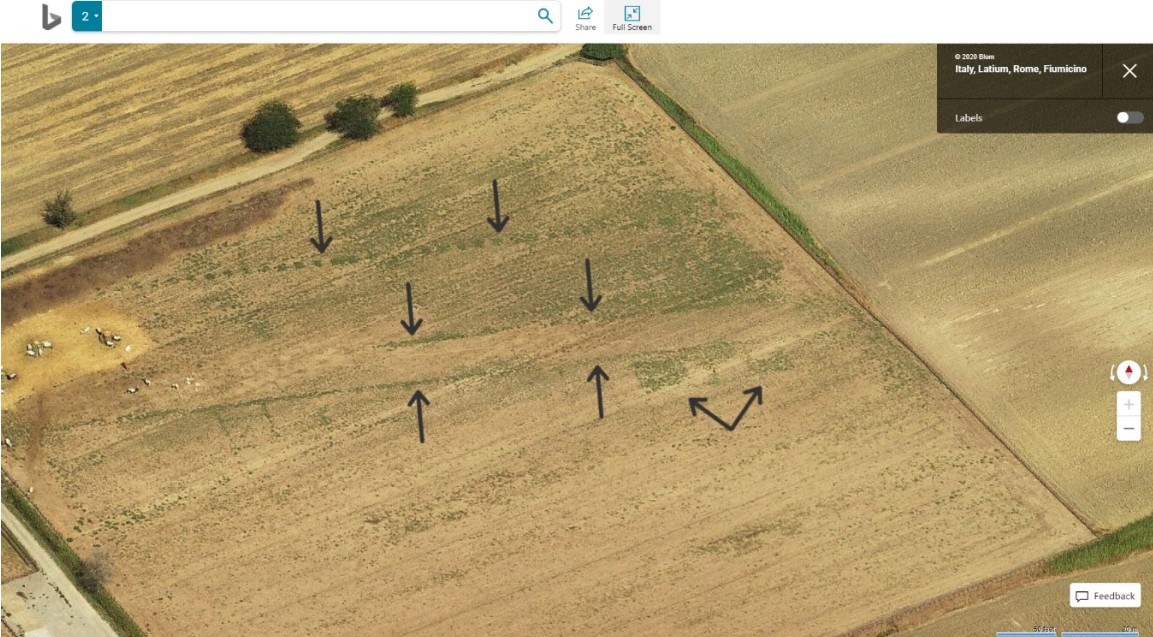

**Figure 10.** An example of archaeological cropmarks with one positive detection. These structures are similar to those identified by an extensive magnetometry survey published by Keay et al. in 2005 [52]. In this survey, the dotted lines traversing east to west have been interpreted as an aqueduct and the lines parallel to the dotted line as a gravel road, both traversing from the hexagonal port of the Roman emperor Trajan to the Tiber river. The cropmarks below the road have been interpreted as various constructions. The black arrows point to some of these features. The Microsoft product screenshot is reprinted with permission from Microsoft Corporation.

The possibility exists that tasks with zero positive answers may still have included archaeological cropmarks. Of the 17,251 tasks classified with zero detections as of 5 July 2020, 500 were checked by a local expert from the Tor Vergata University of Rome, with both knowledge of the local area and with expertise in archaeological interpretation of remote sensing data. None of these tasks were deemed by this interpretation to contain unambiguous archaeological cropmarks. However, this is only a small fraction of the total tasks completed with zero detections, and the presence of archaeological cropmarks, or lack thereof, cannot be ascertained with complete certainty without supporting information or ground survey. It is therefore difficult to quantify possible errors of omission.

## 4. Discussion

While the tasks are only 28% completed, at least one full coverage of the AOI has undergone human interpretation at least once. The results so far would suggest that interpretation of remotely sensed data by citizen researchers may facilitate archaeological surveys through the identification of potential archaeological residues in large amounts of data. However, any discussion of the accuracy of this approach should be treated with caution. Crowdsourcing may help retrieve the odd cropmark that may otherwise go undetected, but it is unlikely to replace systematic archaeological surveys. While a sample of tasks with zero detections was checked, a thorough assessment of possible false negatives would require checking all tasks. This would have been too time-consuming, and would have defeated the purpose of using citizen science to alleviate the burden of interpretation. Moreover, remotely sensed data alone is often not enough to carry out archaeological interpretation, even by experts. Often more information is required to reduce ambiguities, such as local knowledge, ground survey, or data from other sensors. Another aspect to consider is that a significant quantity of archaeological remains in the study area are from the Roman period. In this age, structures such as roads, urban areas, and building foundations are generally characterized by geometric shapes formed by straight lines. Cropmarks of these features may be easier to interpret by non-expert analysis than structures from certain other periods. Possible biases due to this must also be taken into account.

Archaeological prospection raises ethical considerations. While the project is based on data that is in any case available to the public, encouraging citizen scientists to detect archaeological structures may attract looters. No location information was provided with the tiles in each task, but nonetheless, the possibility exists that through use of the platform, a potential looter may be made aware of the existence of archaeological sites, and of the potential lack of protection surrounding it, and attempt to seek more information. This could be mitigated by inviting participants only from a selected community of users, such as heritage practitioners. It would reduce the number of users, but may increase the quality of interpretation if such a community is better trained for archaeological analysis of remote sensing data.

The completion of tasks is highly dependent on the task definition, quantity of tasks, and on the success in project promotion [60]. Even if one full coverage was successfully completed, it may be that cropmarks in a given area are only apparent in another Bird's Eye acquisition acquired in one (or more) of the other four orientations. For successful completion of the project, many more tasks are therefore still to be completed, and ideally with a certain redundancy. The use of gamification and other techniques [60] to attract users and retain them may have facilitated the completion of these tasks. Some of these may be adopted in a future version of the platform. Another level of citizen engagement could involve Volunteered Geographic Information (VGI) [61–63] to quality control cropmark detections using local knowledge of citizen researchers. However, this would involve sharing location information on potential detections and may raise ethical concerns, as discussed above.

Upon the completion of all tasks, if a critical mass of positive detections has been reached, the next phase of the project may assess the feasibility of using these to train a machine-learning algorithm to recognise cropmarks in similar data, such as that carried out by [47]. However, there is an enormous variety in the form that cropmarks can take, even if limited to a certain object type and period [17]. Significant variety also exists in the clarification of cropmarks and the conditions under which they

may appear [32,34,64]. Many training samples are likely to be needed for a machine learning model to be successful and inventive data augmentation techniques to expand the available samples may be fundamental.

Another future activity would consider the use of other remotely sensed data of detected cropmarks to compare performance in cropmark discrimination, such as VHR SAR, hyperspectral, LiDAR and other types. However, due to the ephemeral nature of many cropmarks, a fair comparison is difficult to make. Even if similar acquisition dates are available for different data types, suitable conditions for cropmark appearance may vary according to the type (e.g., SAR is more sensitive to moisture and roughness signatures of cropmarks [15], while optical data reveal colour variations [17,32,33]). Frequently acquired data from the Copernicus missions, especially Sentinel-2 optical imagery, may be useful in this respect, although the spatial resolution (max 10 m for Sentinel-2) is a limitation. Nonetheless, a future activity could assess the feasibility of applying techniques, such as those based on machine learning, to recognise cropmarks in lower resolution data.

## 5. Conclusions

This pilot project is a proof of concept of the potential of crowdsourcing for the detection of archaeological cropmarks in the heritage-rich area surrounding the city of Rome, Italy. A number of positive cropmark detections have been verified with available results of ground surveys. These results demonstrate that the Bird's Eye service of Bing Maps contains a wealth of information that could be useful for archaeological survey. It is also an encouraging example of community empowerment and a participatory approach to heritage.

However, it is important not to oversell such an approach, particularly given that an exhaustive accuracy assessment of potential false negatives has not been carried out, and that proper interpretation of potential archaeological cropmarks usually requires other data. Archaeological cropmarks vary greatly in complexity. Cropmarks from the Roman period in the area surrounding the modern city of Rome may be easier to interpret than more challenging archaeological residues in other locations and from other historical periods. It remains to be seen whether the results obtained here would be comparable to other locations. Moreover, the use of an open platform for archaeological prospection raises ethical considerations.

The work presented here is a description and preliminary assessment of a project that is 28% completed. Improvements could be made to speed up the completion of tasks through better promotion and use of techniques such as gamification to retain users. If a critical mass of positive detections of cropmarks is reached, this data could be used to train a machine-learning algorithm to carry out systematic prospection on a larger scale.

**Author Contributions:** Conceptualization, C.S. and G.L.; methodology, G.L., C.S. and D.L.G.; software, G.L. and D.L.G.; validation, C.S.; formal analysis, C.S.; investigation, C.S.; writing—original draft preparation, C.S.; writing—review and editing, G.L.; project administration, D.L.G. All authors have read and agreed to the published version of the manuscript.

**Funding:** This research received no external funding.

**Acknowledgments:** We would like to extend a huge 'thanks' to all the citizen researchers, without whom this activity would not have been possible.

**Conflicts of Interest:** The authors declare no conflict of interest.

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
