# Peer review of "A Pilot Study on Remote Sensing and Citizen Science for Archaeological Prospection"

_remotesensing, doi:10.3390/rs12172795_

Round 1

Reviewer 1 Report

This paper deals with one of the grand challenges for archaeology and remote sensing in archaeology: the use of crowdsourcing for the detection of unknown archaeological sites.

The authors  evaluate the potential to use the Microsoft Bing Maps Bird’s Eye service within a crowdsourcing platform to identify archaeological proxy indicators (crop-marks) in the surroundings of the city of Rome in Italy

The article is well written, the approach is correct, the results are interesting

Some minor revisions are requested

1_provide additional details on the phenomenology of the most common proxies (crop/soil marks, moisture patterns)

2_Ethic issue related to crowdsourcing as a mean to detect archaeological features

The authors say that "no location information was provided with the tasks .." so " it is unlikely that looting was the main incentive of users", well...I think that a more in-depth analysis of the topic needs to be done.

I suggest providing additional consideration about the ethic issue: this kind of activity (if the data are available)  could encourage looting activity. What are the precautions to prevent data from getting hold of everyone?

                3_ Methods.

Summarize the methodological approach in a graphic scheme or a flowchart including the diverse work phases and their connections

4

"Clearly the higher the redundancy, the higher the accuracy": could the authors provide additional data about the relation between redundancy and accuracy?

5

Conclusion: Move the future perspectives of this research in the conclusion

Reference suggestions:

About crowdsourcing and archaeology see

Kintigh, K.W. et al. 2014, Grand challenges for archaeology, Proceedings of the National Academy of Sciences Jan 2014, 111 (3) 879-880; DOI: 10.1073/pnas.1324000111

Kintigh, K.W. (2013) Grand Challenges for Archaeology: Crowd Sourcing Report. https://core.tdar.org/document/391233

About crop marks as proxy indicators using SAR image see

Jiang A., Chen F. et al. (2016), Archeological crop marks identified from Cosmo-SkyMed time series: the case of Han-Wei capital city, Luoyang, China. International Journal of Digital Earth, 10(8), 846-860, http://dx.doi.org/10.1080/17538947.2016.1254686

About crop-marks from satellite multispectral imagery and their seasonality

Agapiou A., Lysandrou V. et al. (2016). Study of the Variations of Archaeological Marks at Neolithic Site of Lucera, Italy Using High-Resolution Multispectral Datasets. Remote Sens. 2016, 8, 723; doi:10.3390/rs8090723

Author Response

1_provide additional details on the phenomenology of the most common proxies (crop/soil marks, moisture patterns)

This has now been included. Please see lines 64 to 91.

2_Ethic issue related to crowdsourcing as a mean to detect archaeological features

The authors say that "no location information was provided with the tasks .." so " it is unlikely that looting was the main incentive of users", well...I think that a more in-depth analysis of the topic needs to be done.

I suggest providing additional consideration about the ethic issue: this kind of activity (if the data are available)  could encourage looting activity. What are the precautions to prevent data from getting hold of everyone?

A better and more thorough discussion is now provided on this topic, please see lines 414 to 422.

                3_ Methods.

Summarize the methodological approach in a graphic scheme or a flowchart including the diverse work phases and their connections

A new figure has been added that summarises the method. See figure 3.

4

"Clearly the higher the redundancy, the higher the accuracy": could the authors provide additional data about the relation between redundancy and accuracy?

 The article has been extensively revised and no longer makes a connection between redundancy and accuracy in the context of this activity.

5

Conclusion: Move the future perspectives of this research in the conclusion

Another reviewer suggested no new information be added in the conclusion. The discussion section refers to future activities in detail, with a summary in the conclusions section.

Reference suggestions:

About crowdsourcing and archaeology see

Kintigh, K.W. et al. 2014, Grand challenges for archaeology, Proceedings of the National Academy of Sciences Jan 2014, 111 (3) 879-880; DOI: 10.1073/pnas.1324000111

Kintigh, K.W. (2013) Grand Challenges for Archaeology: Crowd Sourcing Report. https://core.tdar.org/document/391233

About crop marks as proxy indicators using SAR image see

Jiang A., Chen F. et al. (2016), Archeological crop marks identified from Cosmo-SkyMed time series: the case of Han-Wei capital city, Luoyang, China. International Journal of Digital Earth, 10(8), 846-860, http://dx.doi.org/10.1080/17538947.2016.1254686

About crop-marks from satellite multispectral imagery and their seasonality

Agapiou A., Lysandrou V. et al. (2016). Study of the Variations of Archaeological Marks at Neolithic Site of Lucera, Italy Using High-Resolution Multispectral Datasets. Remote Sens. 2016, 8, 723; doi:10.3390/rs8090723

Thank you for these suggestions, some have been included.

Reviewer 2 Report

The authors present an interesting topic dealing with the detection of crop marks based on crowdsource information. While the findings presented here, are part of an ongoing project, interesting findings are there. Before publication i would reccomend the authors to make the following changes so as to improve their work and provide more information to readers.

  1. In general a wider introdcution would be appreciated. One-two paragraphs dealing with the use of optical remote sensing for the detection of crop marks could help readers to better understand the problems through interpretation.
  2. More details regarding the platform are needed. This should be supported by figures. At the moment the readers can only view Bing screenshots and not the "system" itself
  3. Please elaborated further the results. More statics would be very helpful, eg. time of interpretation, with graphs (e.g. median, mean, std time etc). 
  4. Some examples from FP and TP would be also supportive.
  5. Conclusion section should be expanded
  6. Please also elaborate the policy of the Bing platform. is this an open source information? the bird's eye view is not applicable in all areas, but only in specific areas of interest. Please make clear that these data are aerial not satellite.
  7. How do you see platform to work in the  near future? 
  8. Probable the title can be more specific.

I think the authors can address all of these points and improve the paper.

Author Response

1. In general a wider introdcution would be appreciated. One-two paragraphs dealing with the use of optical remote sensing for the detection of crop marks could help readers to better understand the problems through interpretation.

This has now been provided. Please see lines 72 to 81 (and revised paragraphs before and after as well).

2. More details regarding the platform are needed. This should be supported by figures. At the moment the readers can only view Bing screenshots and not the "system" itself

The methodology section has been revised and two new figures describe the platform (please see figures 3 and 6).

3. Please elaborated further the results. More statics would be very helpful, eg. time of interpretation, with graphs (e.g. median, mean, std time etc).

The results have been discussed in more detail in the discussion section. Unfortunately it was not possible to add reliable statistics of users given that the project took place in two phases (initially on Crowdcrafting, then on dedicated ESA platform), and some statistics were lost. The new platform only recorded recent activity. However, in the discussion section, possible future improvements are mentioned.

4. Some examples from FP and TP would be also supportive.

We did not want to insert too many screenshots, and there were so many false positives that it would have been difficult to select a representative sample. Nonetheless, examples of true positives are included as figures.

5. Conclusion section should be expanded

The conclusion section has now been expanded to include a summary of the successes and limitations of the project, and future improvements and activities.

6. Please also elaborate the policy of the Bing platform. is this an open source information? the bird's eye view is not applicable in all areas, but only in specific areas of interest. Please make clear that these data are aerial not satellite.

The policy of Bing Maps is now stated (free service, but Bird’s Eye data can only be visualised). The data coverage is also mentioned, as is the fact that the data are aerial not satellite.

7. How do you see platform to work in the  near future? 

Future activities are now described in the discussion and results section.

8. Probable the title can be more specific.

The title has been modified. It is now a lot more specific: “A Pilot Study on Remote Sensing and Citizen Science for Archaeological Prospection”

Reviewer 3 Report

Dear Authors,

Thank you very much for the interesting article about the usability of citizen science for detecting crop marks.
I do have remarks / comments / reservations about the article. See the attached file for the more specific comments (textual etc.). I hope the comments in the file are clear.
In the following some general remarks:

General: I prefer the term citizen science instead of crowdsourcing. To me crowdsourcing has a financial element connected to it. But this is a matter of taste. The same goes for the term used for the users, I prefer citizen researchers. 
I think that when you refer to a publication in Remote Sensing you just add the number not the name: as identified by Stewart [25].

Title: I think the title of the article is too generic and does not cover the content of the article. When I first saw the title I expected a review of the use of citizen science in archaeological prospection. Instead I would consider this a pilot project in the usability of citizen science for detecting archaeological crop marks. The title needs to be more specific and should maybe also implicate that these are preliminary results. 

Method / Results: My main reservation with the article lies in the methodology. While I understand the change made in task redundancy (it is unfortunate that the original platform ceased to exist) this, in my opinion results in a serious problem. The main problem is that confidence thresholds cannot be used any more and the number of positive answers says nothing about the confidence of the users / image but simply how many users have seen it. If I understand correctly a tile within an image can possibly be in 1-8 different images (orientation and overlap), which can be seen by 1-3 users. As this is not clear for every tile, and the redundancy varies during the project, it becomes impossible to put any value to the number of detections. I would implore the authors to revise and reconsider the results and present them more as a confirmation of the hypotheses that citizen science can be used for crop mark detection. Without a full coverage by an equal number of users, and insight in the possible views per tile, confidence thresholds cannot be used in my opinion.
For instance, the results presented in line 242-243 are not surprising as the majority of images is only seen by one user! pooling all positive detections would be more appropriate.

References: The references section needs to be seriously adjusted. I am missing most editors of volumes, all DOIs, and cities for books. Please revise the references.   

If you would consider changing the focus of the article to a pilot study or proof of concept that shows the usability of CS for crop mark detection I think the article would be better, especially because the presented results are preliminary. This would also be research that adds to the ongoing study into the use of CS in archaeological remote sensing. 

Author Response

General: I prefer the term citizen science instead of crowdsourcing. To me crowdsourcing has a financial element connected to it. But this is a matter of taste. The same goes for the term used for the users, I prefer citizen researchers. 

The terms citizen science and citizen researchers are now used throughout the manuscript.

I think that when you refer to a publication in Remote Sensing you just add the number not the name: as identified by Stewart [25].

This has been adopted for all references. Only on the occasions when a specific name or institution is referred to is it still included.

Title: I think the title of the article is too generic and does not cover the content of the article. When I first saw the title I expected a review of the use of citizen science in archaeological prospection. Instead I would consider this a pilot project in the usability of citizen science for detecting archaeological crop marks. The title needs to be more specific and should maybe also implicate that these are preliminary results. 

The title has been changed to make it more specific to the activities of this research: “A Pilot Study on Remote Sensing and Citizen Science for Archaeological Prospection”

Method / Results: My main reservation with the article lies in the methodology. While I understand the change made in task redundancy (it is unfortunate that the original platform ceased to exist) this, in my opinion results in a serious problem. The main problem is that confidence thresholds cannot be used any more and the number of positive answers says nothing about the confidence of the users / image but simply how many users have seen it. If I understand correctly a tile within an image can possibly be in 1-8 different images (orientation and overlap), which can be seen by 1-3 users. As this is not clear for every tile, and the redundancy varies during the project, it becomes impossible to put any value to the number of detections. I would implore the authors to revise and reconsider the results and present them more as a confirmation of the hypotheses that citizen science can be used for crop mark detection. Without a full coverage by an equal number of users, and insight in the possible views per tile, confidence thresholds cannot be used in my opinion.
For instance, the results presented in line 242-243 are not surprising as the majority of images is only seen by one user! pooling all positive detections would be more appropriate.

Thank you very much for this insight. We fully agree and have extensively revised the manuscript to present the research as a pilot study on citizen science for archaeological prospection with remote sensing, without making any claims on confidence levels or accuracy. The results, discussion and conclusions present some positive results but stress caution given the lack of accuracy assessment.

References: The references section needs to be seriously adjusted. I am missing most editors of volumes, all DOIs, and cities for books. Please revise the references.   

The references have been thoroughly reviewed, with missing information added, wherever possible. DOIs have also been added to most references.

If you would consider changing the focus of the article to a pilot study or proof of concept that shows the usability of CS for crop mark detection I think the article would be better, especially because the presented results are preliminary. This would also be research that adds to the ongoing study into the use of CS in archaeological remote sensing. 

The article has been extensively revised taking into account the comments listed here, and those reported in the pdf. The article focus has now been shifted to a report of preliminary results of a pilot study on the use of citizen science for archaeological prospection using the Microsoft Bing Maps Bird’s Eye WMS. Rather than attempting to discuss accuracy and confidence of detections, the article merely reports the results obtained and discusses the potential utility of these, emphasising the advantages and limitations of the adopted approach. All the comments were extremely constructive, and any further suggestions are very much appreciated.

Reviewer 4 Report

The manuscript deals with an interesting topics about the crowdsourcing of satellite images to detect archeological signatures. Particularly, the manuscript highlights the results on a specific area of a preliminary calibration of the available platform. The final goal is to check if non-expert interpretation of archaeological crop marks in remotely sensed VHR optical imagery through crowdsourcing could facilitate systematic expert analysis, as a first filter to detect possible archaeological crop marks over wide areas.

In such a context, even if the topic looks interesting, I have some concerns about the scientific level, description of the method, and calibration of the method itself.

The section 2.2 deals with generic technical issues of the platform that could be condensed, meanwhile the scientific bachground on the crop-marks and recognition of archaelogical issue is missed; also details on the tools that users can applied for improving the detectability on the images.

The authors  generically statethat some tools are available for the detection, probably they concern with filters or edge detection tools. It could be interesting to assess the most adopted tools used, the flow chart the users have adopted and the tools that have proven most successful. A more sistematic overview of the system could be of interest. 

The meaning of different orientation of the image and how the interpreters can use the different orientations require a detailed analysis. The advantages of the oblique image with respect of other images will require an explaination. The resolution of the available images with respect to the exepcted size of the archaelogical remains is also missing. 

The calibration of the method is also not well detailed. Authors just report some images without any clear indication of the detected signatures, and without any comparison with previoulsy detections performed using other methods. Therefore, the results section must be improved, including a clear indication of the archeological evidences in the selected examples (in the figures) and a more detailed comparison with data/images of previous studies. 

About the task of  false negative, authors states that a sample of 500 completed task have been interpreted by an expert user, and none were deemed to contain missed detections. What does it mean  expert user ? How they can prove the reliability of the expert user ? 

Moreover, it must be fixed in advance the meaning of archeological remains: the user (but also the expert involved in the calibration) could be biased in this sense. In the selected area, the main features are probably the remains of villas and villages, that in the romain age they were well articulated from a geometric point of view. But other archeological targets could be expected in the region such as pre-romaninc remains (for instance neolitic tombs, potteries, fires and so on). This implies some preliminar evaluation on the detectability of other not so well organised remains, and implies also a minimal "education" and knowledge and skills of the users in the different kind of archeolgical remains and in searching method of archeological evidences.

The manucript looks more a report on the preliminary results of the project than a scientific paper. I encourage the authors to check it carefull and to emphasize the scientific background of the project. For instance one possible task that could be improved is the relevance of the redundancy of the images and how this redundancy could facilitate the non-expert interpretation.

Author Response

The section 2.2 deals with generic technical issues of the platform that could be condensed, meanwhile the scientific bachground on the crop-marks and recognition of archaelogical issue is missed; also details on the tools that users can applied for improving the detectability on the images.

The introduction has been extensively developed to include scientific background of cropmarks and use of tools and processing techniques for their recognition (see lines 59 to 125).

The authors  generically statethat some tools are available for the detection, probably they concern with filters or edge detection tools. It could be interesting to assess the most adopted tools used, the flow chart the users have adopted and the tools that have proven most successful. A more sistematic overview of the system could be of interest. 

A discussion of such tools has been included (see 108 to 115). Also, a flow chart of the various processes and functions of the project has been included (see figure 3), together with a more systematic overview of the research (see last part of introduction and various modifications to section 2).

The meaning of different orientation of the image and how the interpreters can use the different orientations require a detailed analysis. The advantages of the oblique image with respect of other images will require an explaination. The resolution of the available images with respect to the exepcted size of the archaelogical remains is also missing. 

The utility of the different orientations and oblique angle of acquisition is now discussed, as has the resolution of the available images with respect to the size of the archaeological remains (see lines 117 to 125, and caption of figure 4).

The calibration of the method is also not well detailed. Authors just report some images without any clear indication of the detected signatures, and without any comparison with previoulsy detections performed using other methods. Therefore, the results section must be improved, including a clear indication of the archeological evidences in the selected examples (in the figures) and a more detailed comparison with data/images of previous studies. 

Results are compared, where possible, with results of other surveys, including geophysical survey of Portus by Kaey, magnetometry survey of Veii by Campana, and results of previous surveys of the Via Salaria by Quilici, Hypponen and others. However, the actual results of these surveys are not included in figures due to copyright issues, but the references are given.

About the task of  false negative, authors states that a sample of 500 completed task have been interpreted by an expert user, and none were deemed to contain missed detections. What does it mean  expert user ? How they can prove the reliability of the expert user ? 

The expert user is from the University of Rome Tor Vergata with both knowledge of the local area and with expertise in archaeological interpretation of remote sensing data. This is now mentioned in the manuscript. It is also mentioned that the presence of archaeological cropmarks, or lack thereof, cannot be ascertained with complete certainty without supporting information, or ground survey. It is therefore difficult to quantify possible errors of omission. (See lines 378 to 386, and parts of abstract, discussion and conclusions.)

Moreover, it must be fixed in advance the meaning of archeological remains: the user (but also the expert involved in the calibration) could be biased in this sense. In the selected area, the main features are probably the remains of villas and villages, that in the romain age they were well articulated from a geometric point of view. But other archeological targets could be expected in the region such as pre-romaninc remains (for instance neolitic tombs, potteries, fires and so on). This implies some preliminar evaluation on the detectability of other not so well organised remains, and implies also a minimal "education" and knowledge and skills of the users in the different kind of archeolgical remains and in searching method of archeological evidences.

This is a very valid point. It has now been discussed more thoroughly in lines 399 to 404.

The manucript looks more a report on the preliminary results of the project than a scientific paper. I encourage the authors to check it carefull and to emphasize the scientific background of the project. For instance one possible task that could be improved is the relevance of the redundancy of the images and how this redundancy could facilitate the non-expert interpretation.

The manuscript has been extensively revised following this comment, and that of other reviewers. The focus has been shifted to emphasize that this is a preliminary report of a pilot activity. The scientific background has been developed. Also, the discussion on redundancy has been completely revised, and is no longer associated with confidence levels.

Round 2

Reviewer 1 Report

All the corrections were punctually made, comments and suggestions encouraged a revision which significantly improved the manuscript which is now ready for publication

Author Response

Thank you very much for your review. Please find attached a modified version of the manuscript with some minor corrections.

Reviewer 2 Report

The authors have revised their paper, and i think it worths for publication. It is of great importance that such crowd-sourced investigations for promoting space archaeology have attracted the interest of the scholar (and the broader public as end-users of the system). However, at the same time the authors address and highlight some critical aspects (ethics) of such efforts: is the missing of the geolocation of the images enough to prevent looting or is this crowd-sourced platform helpful for assisting experts in the field.

Following this, I would like to see the overall results of the platform and the lesson learned from this experience.

Author Response

(The authors gave the same response as above.)

Reviewer 3 Report

Dear Authors,

Thank you for adjusting the article as per my recommendations. I think the article is greatly improved. 

In the attached file a few minor typos and comments. 

Author Response

Many thanks for those comments. We have inserted a reference where requested, and corrected the various typos.

Reviewer 4 Report

Dear Authors,

the new release considers the suggestions and comments of the previous release and the overall manuscript has been improved. I dont have any other issue and comment on the paper.

Sincerely

Author Response

(The authors gave the same response as above.)
